# NIGWO-iCaps NN: A Method for the Fault Diagnosis of Fiber Optic Gyroscopes Based on Capsule Neural Networks

**DOI:** 10.3390/mi16010073

**Published:** 2025-01-10

**Authors:** Nan Lu, Huaqiang Zhang, Chunmei Dong, Hongtao Li, Yu Chen

**Affiliations:** 1Department of Instrumentation, School of Mechanical Engineering, West Campus, Shandong University of Technology, Zibo 255000, China; lunan20001125@163.com (N.L.); dcmjob@126.com (C.D.); myyolo@163.com (H.L.); 2Beijing Institute of Space Launch Technology, Beijing 100076, China; chenyu@aspe.buaa.edu.cn

**Keywords:** fiber optic gyroscope, fault diagnosis, capsule neural network, gray wolf algorithm

## Abstract

When using a fiber optic gyroscope as the core measurement element in an inertial navigation system, its work stability and reliability directly affect the accuracy of the navigation system. The modeling and fault diagnosis of the gyroscope is of great significance in ensuring the high accuracy and long endurance of the inertial system. Traditional diagnostic models often encounter challenges in terms of reliability and accuracy, for example, difficulties in feature extraction, high computational cost, and long training time. To address these challenges, this paper proposes a new fault diagnostic model that performs a fault diagnosis of gyroscopes using the enhanced capsule neural network (iCaps NN) optimized by the improved gray wolf algorithm (NIGWO). The wavelet packet transform (WPT) is used to construct a two-dimensional feature vector matrix, and the deep feature extraction module (DFE) is added to extract deep-level information to maximize the fault features. Then, an improved gray wolf algorithm combined with the adaptive algorithm (Adam) is proposed to determine the optimal values of the model parameters, which improves the optimization performance. The dynamic routing mechanism is utilized to greatly reduce the model training time. In this paper, effectiveness experiments were carried out on the simulation dataset and real dataset, respectively; the diagnostic accuracy of the fault diagnosis method in this paper reached 99.41% on the simulation dataset; the loss value in the real dataset converged to 0.005 with the increase in the number of iterations; and the average diagnostic accuracy converged to 95.42%. The results show that the diagnostic accuracy of the NIGWO-iCaps NN model proposed in this paper is improved by 13.51% compared with the traditional diagnostic methods. It effectively confirms that the method in this paper is capable of efficient and accurate fault diagnosis of FOG and has strong generalization ability.

## 1. Introduction

Fault diagnosis and isolation play a crucial role in monitoring system stability and reliability, and the diagnostic process can usually be divided into three processes: signal processing, feature extraction, and fault identification [1,2]. An inertial navigation system (INS) relies on gyroscopes to provide accurate attitude angles, angular velocities, and accelerations [2,3]. INS can be affected by external environmental conditions such as weather, spoofing attacks, magnetic interference, and jammers [4,5,6], which may lead to the malfunctioning of the inertial measurement unit (IMU) [7]. Fiber optic gyroscopes are widely used in aerospace, marine, automotive, robotics, and other high-precision measurement fields as the core of inertial measurement units. In navigation and control systems, gyroscope failures can cause system failure with irreversible, severe consequences, which can be fatal for military, satellite, and missile applications. The timely and effective troubleshooting of fiber optic gyroscopes can promptly identify potential problems and help the system identify and repair factors affecting the accuracy of navigation measurements. With the development of the Internet of Things and big data technology, fault diagnosis and data analysis can achieve intelligent monitoring and fault prediction of the fiber optic gyroscope, thus improving the system’s intelligence. Gyroscope fault diagnosis is mainly based on signal processing, model-based, and artificial intelligence-based methods. For complex faults, signal processing-based fault diagnosis methods make it difficult to extract feature information with sufficient differentiation, and in industrial environments, they are sensitive to noise, which affects the accuracy of feature extraction. For example, Gu et al. proposed a fault identification algorithm combining the signal processing algorithm and the machine learning algorithm for four-mass vibration MEMS gyroscopescopes [8]. However, when using a machine learning algorithm to extract features, the spatial relationship between different features will be lost, making feature extraction inadequate. In this paper, the capsule neural network algorithm is proposed to avoid this deficiency. Model-based diagnostic methods describe the behavior of a device by constructing a mathematical model of the system (e.g., a physical model, a state space model, a fault model, etc.) and analyzing it in comparison with the actual measured data. Common methods include Kalman filtering, state estimation, and fault detection models. For simple systems, this is a good choice. However, for complex or nonlinear systems, it is very difficult to establish an accurate mathematical model without access to sufficient system information, and this method requires very powerful computational power and high computational complexity. For example, Bai et al. proposed a new method to diagnose laser gyroscope faults using a kernel-limit learning machine, which effectively remedies the defects of the traditional diagnostic model, such as harrowing feature extraction and high technical cost [9]. For large datasets, the extreme learning machine will cause memory overflow when calculating and storing the kernel matrix, and the application scope will be very limited. Qian et al. proposed an intelligent method combining wavelet transform and RBF neural networks to improve the efficiency of fault diagnosis [10]. However, the performance of the RBF network is highly dependent on the choice of radial basis function center. A central point determined by human experience can cause network performance to degrade. In this paper, the parameters of the capsule neural network are determined by combining the optimized gray wolf algorithm with the Adam algorithm. This avoids deficiencies due to human defects. To maintain the system with a long-range and high accuracy during navigation, troubleshooting interferometric fiber optic gyroscopes (IFOGs) is critical to studying the soft failures of the INS [11,12,13].

As a typical representative of deep learning, convolutional neural networks (CNNs) use maximum pooling to accelerate significantly the convergence and broaden the sensory field [14]. However, training a model requires many samples [15], and the pooling process loses the deep spatial relationships between different features [16,17]. The capsule neural network (Caps NN), proposed by G.E. Hinton in 2011 [18,19], avoids the adverse effects caused by maximum pooling well and improves the feature extraction ability of the model. For example, Cao et al. [20] proposed a dual augmented capsule neural network using perceptual attention for face recognition, which enhances the ability of feature representation; Zhu et al. proposed a capsule network with starting blocks and regression branches for fault diagnosis of bearings, which verifies the generalization performance of capsule networks in fault diagnosis [17]. The one-dimensional attentional convolutional capsule neural network proposed by Ye et al. [21], through the introduction of capsule layers, ensures the completeness of fault feature extraction and achieves a more desirable accuracy rate with a small number of samples. In recent years, Caps NN has been widely applied in fault diagnosis. However, the application of Caps NN is still relatively small in the fault diagnosis of inertial navigation.

Aiming at IFOGs’ problem of lacking apparent fault feature data and the uncertainty of human experience in fault diagnosis, this paper constructs the NIGWO-iCaps NN (Zibo, China), which is the iCaps NN optimized by the NIGWO. Based on the traditional capsule neural network, the features among the data are extracted deeply through the addition of a deep feature extraction module (DFE). The combination of the NIGWO and the Adam algorithm can avoid gradient vanishing while greatly improving diagnostic efficiency.

The rest of the paper is structured as follows. Section 2 is the acquisition and pre-processing of IFOG fault data. Section 2.1 is the establishment of the simulation model of the fiber optic gyroscope, which mainly contains two parts, the dynamic model and the stochastic model; Section 2.2 introduces how to build the fault model of IFOG in detail. Section 2.3 introduces the process of constructing the feature matrix. We use wavelet packet transform to extract the feature of the signal and obtain the feature matrix through the conservation law of the energy of the signal and the wavelet packet reconstruction coefficient. Section 3 is the design and implementation of the NIGWO-iCaps NN model. It mainly includes the optimization of the gray wolf algorithm, the improvement of the capsule neural network, and the overall diagnosis process design. Section 4 is the experiments, which clarifies the evaluation indexes and defines the parameters of the model; and Section 5 is the experimental results and discussion. We compared the simulation data and actual data respectively. The experimental results show the high diagnostic accuracy and strong generalization ability of the model in this paper. Section 6 is the conclusion.

## 2. Acquisition and Pre-Processing of IFOG Fault Data

### 2.1. Building the Simulation Model of IFOGs

Aiming at the problems of the fault signals of IFOGs being difficult to obtain in large quantities and the characteristics of the actual fault signals being not obvious, this paper generates the fault signals by establishing a simulation model of the fault data of IFOG, which includes two parts, namely dynamic modeling and stochastic modeling.

The dynamic modeling of IFOGs is based on the common digital closed-loop fiber optic gyroscope to establish a linear discrete model. In the digital closed-loop fiber optic gyroscope system, the connection between the Sagnac phase shift $\varphi_{c}$ and input angular velocity ω is realized based on the Sagnac effect [22].

The light emitted from the laser beam is divided by the beam splitter into two coherent beams propagating in reverse by the splitter. When the fiber ring rotates at the angular speed ω, the interference effect occurs after the two beams pass through a complete rotation path, resulting in a phase difference $\varphi_{c}$, known as the Sagnac phase shift. By measuring the differential phase shift, we convert the rotation of the angle into an offset of the interference pattern of the two beams [23].

Figure 1 shows the mechanism of the Sagnac effect. It is assumed that point A on the optical fiber ring is the initial injection point of the light wave, which divides into two beams after entering the optical fiber ring. The two beams of light travel in opposite directions at the same time. Set the light beam CCW propagating clockwise and CW propagating counterclockwise. The propagation speed of light in vacuum is c, and the refractive index of the material used in the fiber ring is n. When the fiber ring does not rotate with respect to the inertial space, the propagation speed of CW and CCW is c/n. If a ring interferometer is given a clockwise rotational angular velocity ω with respect to the inertial reference frame, and the injection point moves to $A^{\prime }$ after time $\tau_{c}$, the paths of the two beams that meet at that point are different. The path of a CCW beam propagating clockwise is(1)$$L_{CCW}=2\pi R+R\omega t_{CCW}=C_{CCW}t_{CCW}$$
where R is the radius of the fiber ring, $t_{CCW}$ is the time that the beam CCW passes through the fiber ring, and $C_{CCW}$ is the propagation speed of the beam CCW in the fiber ring.

A beam CW propagating counterclockwise travels through an optical distance of(2)$$L_{CW}=2\pi R+R\omega t_{CW}=C_{CW}t_{CW}$$
where $t_{CW}$ is the time that the beam CCW passes through the fiber ring, and $C_{CW}$ is the propagation speed of the beam CCW in the fiber ring.

From Equations (1) and (2), the time difference $\tau_{c}$ between the propagation of the two beams of light in the fiber ring is(3)$$\tau_{c}=t_{CCW}-t_{CW}=\frac{2\pi R}{C_{CCW}-R\omega }-\frac{2\pi R}{C_{CW}+R\omega }$$

The speed of rotation is negligible relative to the speed of light. Therefore, there is Equation (4).(4)$$C_{CCW}=C_{CW}=c/n$$

By calculating from Equations (3) and (4)(5)$$\tau_{c}=\frac{4\pi R^{2}}{c^{2}-{\left(R\omega \right)}^{2}}\omega$$

Since c≫Rω, Equation (5) can be reduced to(6)$$\tau_{c}=\frac{4\pi R^{2}}{c^{2}}\omega$$

Thus, the optical range difference between beams CCW and CW injected into point $A^{\prime }$ after one revolution is(7)$$\Delta L=c\tau_{c}=\frac{4\pi R^{2}}{c}\omega$$

The phase difference produced by the corresponding two beams of light at point $A^{\prime }$ is(8)$$\varphi_{c}=\frac{2\pi \Delta L}{\lambda }=\frac{8\pi^{2}R^{2}}{\lambda c}\omega$$
where λ is the wavelength emitted by the light source.

In practice, to maximize the sensitivity of the fiber optic gyroscope, the fiber optic ring is often multi-turn. Let the number of turns of the fiber optic ring be N and the perimeter of the fiber optic ring be L; Equation (8) becomes(9)$$\varphi_{c}=\frac{4\pi NLR}{\lambda c}\omega$$
where K=4πNLR/λc is the fiber optic gyroscope scale factor.

Figure 2 shows the basic optical path of the interferometric fiber optic gyroscope. There is a light source, a beam splitter, lens A, and a fiber optic ring in the main optical path, and lens B and detectors are distributed on both sides of the main optical path [24]. The light emitted from the light source is transmitted and reflected by the beam splitter. A beam of light through the beam splitter passes through lens A, enters the ring, and propagates clockwise, i.e., CCW light, which comes out of the ring and passes through lens B before being collected by the detector through the beam splitter. Another beam of light goes through the beam splitter after reflecting tibia lens B into the ring, that is, the CW light, and finally goes through lens A on the surface of the beam splitter and is reflected and collected. In this way, the phase difference between the two beams of light is transmitted to the photodetector.

After the photodetector receives the Sagnac phase shift, it generates a voltage signal I(t) after photoelectric conversion, which can be expressed as(10)$$I(t)=\frac{I_{0}}{2}(1+\cos\Delta \varphi )=\frac{I_{0}}{2}(1+\cos(\varphi_{c}+\varphi_{b}-\varphi_{m}))$$
where I(t) is the photodetector output signal, $I_{0}$ is the light intensity, $\varphi_{b}$ is the square wave bias signal, the size is ±π/2, and $\varphi_{m}$ is the phase ramp feedback phase shift.

Figure 3 shows the physical model of the digital closed-loop fiber optic gyroscope [25]. The physical model consists of two parts, the optical structure and the circuit structure. The SLD output light wave arrives at the Y waveguide through the coupler, and the light wave is divided into two beams of equal light intensity, CCW and CW, through the Y waveguide. As the external voltage is applied by the rear drive circuit on the Y waveguide electrode, the phase of the passing CCW light wave changes, indirectly affecting the phase difference of the signal. Therefore, the CCW light wave passing through the Y waveguide can realize the modulation of the phase. Then, the two beams of light rotate along the opposite direction for one week in the fiber optic ring, and then there is an injection point $A^{\prime }$ that arrives at the Y waveguide again with the interference effect occurring. A photodetector is connected to one end of the coupler for the detection of the interference signal.

After the output signal of the photodetector I(t) is filtered and amplified, it is demodulated after A/D conversion, and then the angular rate output is obtained by a digital controller. Meanwhile, the angular rate is fed back to the input after subsequent D/A conversion, a rear driver circuit, and a LiNiO_3_ phase modulator to form a closed-loop feedback system [26]. Through the analysis of the signal transmission process, the dynamic model of digital closed-loop fiber optic gyroscope can be expressed, as shown in Figure 4. In this, a square wave bias is introduced to model the bias error in the gyroscope output signal. This can, to a certain extent, analyze and compensate for the errors caused by the long-term drift of the gyroscope sensor itself, ambient temperature changes, manufacturing differences, and other factors. It can also effectively improve the sensitivity of the fiber optic gyroscope and expand the dynamic range of measurement. We consider that some parts of the system have a certain delay and then add a lag link τ. After the feedback link, the model can obtain the output signal of the fiber optic gyroscope quickly and stably.

As the core of the navigation system, the measurement accuracy of the gyroscope is a key factor affecting the performance of the navigation system. In a combined navigation system, gyroscopes are working in vibration conditions for a long time. The internal components of the device (e.g., fiber optics, sensors) undergo small deformations or displacements. These may cause additional noise. In particular, the bending, compression, or stretching of an optical fiber can alter the light propagation characteristics. That affects the measurement results and produces errors that cannot be eliminated [27,28,29]. By establishing a digital closed-loop dynamic model of a fiber optic gyroscope, we can obtain the angular velocity signals under different fault states and perform the preliminary work for the fault diagnosis.

To reduce the computational complexity of the system simulation, different modules of the dynamic model are simplified into different links, and a linear model is established to reduce the complexity of the system simulation. The Sagnac phase shift can be expressed as a proportional link $K_{1}=2\pi LD/\lambda$, and the photodetector and analog amplification and filtering part can be equated as a proportional link $K_{2}$. The sampling, quantization, modulation, and demodulation processes can be combined into a proportional hysteresis link $K_{3}z^{-1}$. The digital controller uses an integral controller $D(z)={\left(1-z^{-1}\right)}^{-1}$. The D/A conversion, post-amplification drive, and LiNiO_3_ phase modulation process can be combined into a proportional differential process $K_{4}(1-z^{-1})$. The hysteresis link is added to simulate the system’s delay effect, denoted by $z^{-k}$, k = 2. A simulation by Matlab R2021a/Simulink yields the simplified digital closed-loop fiber optic gyroscope linear model shown in Figure 5.

The output response of the system for different angular velocity inputs can be obtained through a simplified linear model. The closed-loop transfer function in the closed-loop control loop shown in Figure 5 can be expressed as(11)$$\begin{array}{c} \;\;\;\;\;\;H(z)=\frac{C(z)}{R(z)}=\frac{K_{1}K_{2}K_{3}z^{-1}D(z)}{1+K_{2}K_{3}z^{-1}D(z)K_{4}(1-z^{-1})z^{-2}/\left(1-z^{-1}\right)}\\ =\frac{K_{1}K_{2}K_{3}z^{-1}D(z)}{1+K_{2}K_{3}D(z)K_{4}z^{-3}}\;\;\;\; \end{array}$$

The error transfer function of the model is(12)$$H_{e}(z)=\frac{E(z)}{R(z)}=\frac{K_{1}}{1+K_{2}K_{3}D(z)K_{4}z^{-3}}$$

The steady-state error of the system is(13)$$e_{ss}=\underset{s\to 1}{\lim}(z-1)E(z)=\underset{s\to 1}{\lim}(z-1)H_{e}(z)R(z)$$

In inertial navigation systems, fiber optic sensors are required to respond quickly and accurately to changes in the input angular rate, so the steady-state error $e_{ss}=0$. To obtain the gain of this closed-loop system, a unit step signal is input, which is obtained according to the median theorem(14)$$\begin{array}{ll} \underset{n\to \infty }{\lim}C(n) & =\underset{z\to 1}{\lim}(z-1)H(z)R(z)\\ & =\underset{z\to 1}{\lim}(z-1)\frac{K_{1}K_{2}K_{3}z^{-1}D(z)}{1+K_{2}K_{3}D(z)K_{4}z^{-3}}\frac{1}{1-z^{-1}}\\ & =\frac{K_{1}}{K_{4}} \end{array}$$

The closed-loop gain of the system is $K_{1}/K_{4}$, which is the calibration factor of the gyroscope.

The parameters of a model of fiber optic gyroscope are taken as an example for the simulation study, where L = 450 m, D = 10 cm. The wavelength of the laser beam produced by the laser diode (LD) is 1.5 μm. The channels of the A/D and D/A converters are 12 and 16 bits, respectively. The unit step signal is taken as an input, and the angular velocity output is obtained by the linear model of the digital closed-loop fiber optic gyroscope by calculating Equations (11)–(14), $K_{1}$ = 0.7744, $K_{3}$ = 812, $K_{4}$ = 9.58 × 10^−5^. To simplify the simulation process, we take $K_{2}$ = 1. The step response curve of the Simulink output with the above parameter settings is shown in Figure 6.

From the analysis of the output results, the steady-state error of the system is 0, the rise time of the step response is about 0.25 ms, and the steady-state value of the unit step response is $K_{1}/K_{4}\approx 8083.5$.The steady-state value of the simulation results is generally consistent with the theoretical value calculation. The step response verifies that the simplified dynamic model can output the signal quickly and stably.

In this paper, the output signal of the IFOG is simulated by adding random noise. When a random input is applied, the output signal of the IFOG random model can characterize the drift characteristics of the gyroscope.

Figure 7 is the stochastic model of a digital closed-loop fiber optic gyroscope. Five kinds of stochastic noises are typically included in an IFOG: angular random wandering noise, zero-bias instability noise, rate random wandering noise, rate ramp noise, and quantization noise [30,31]. The white noise is mainly angular random wandering noise WN_1_, which is simulated by Gaussian white noise with zero mean and variance $\sigma_{1}^{2}$. The colored noise mainly consists of zero-bias instability noise B using 1/f noise, rate random wandering noise WN_2_, rate ramp noise Ramp, and quantization noise QN. WN_2_ is obtained by integrating the Gaussian white noise with zero mean and variance $\sigma_{2}^{2}$. Ramp is simulated by the random ramp function. Ramp is modeled by the random ramp function; QN is modeled by Gaussian white noise with mean zero and variance $LSB^{2}/2$. WN_2_ is integrated to form a fusion noise with WN1, B, and Ramp, and then integrated to accumulate with the quantization noise QN, and finally, the random noise is obtained after differentiation. The article simulates various random noises using a digital signal processing technique. For the stochastic model of the IFOG, it can be expressed as(15)$$\rho (t)=\rho_{0}+K\cos(2\pi ft+\theta_{0})+\sigma \omega (t)+\tau R(t)$$
where ρ(t) is the total output noise of the IFOG, $\rho_{0}$ is the constant value drift of the IFOG, K is the amplitude of the periodic component, σω(t) is the generalized stable white noise with intensity σ, and R(t) is the colored noise.

A tactical-grade FOG was selected for modeling (drift level floated around 0.1 °/h). The parameter settings are as follows:$\;K=3\times 10^{-3}$,  f=100 Hz, $\theta_{0}=\pi /2$,
σ=0.4. Figure 8a shows the output signal of the stochastic model of the IFOG in the normal state.

### 2.2. Establish the Fault Model of IFOGs

IFOGs’ faults include five categories:Bias Fault: When the FOG is subjected to a strong vibration shock, the optical fiber ring is offset or deformed, resulting in a constant deviation between the output signal and the normal signal. The mathematical model is(16)$$y_{m}=\left\{\begin{array}{c} y(t)\\ y(t)+c \end{array}\right.\;\begin{array}{c} t<t_{m}\\ t\geq t_{m} \end{array}$$

2.Drift Fault: This is one of the most common types of faults in FOG. It is usually due to changes in the working environment or internal parameters, such as temperature changes, that the output of the FOG has increased in constant terms. Errors tend to increase over time. The mathematical model is


(17)
$$y_{m}=\left\{\begin{array}{c} y(t)\\ y(t)+kt \end{array}\right.\;\begin{array}{c} t<t_{m}\\ t\geq t_{m} \end{array}$$


3.Blocking Fault: This refers to the point that affects the transmission of the gyroscope signal from a certain point, outputs zero, or maintains the same output value as the previous moment. The mathematical model is


(18)
$$y_{m}=\left\{\begin{array}{c} y(t)\\ y(t_{m}) \end{array}\right.\;\begin{array}{c} t<t_{m}\\ t\geq t_{m} \end{array}$$


4.Periodic Fault: This fault is when the output signal of the FOG starts to attach to the periodically changing signal at some point. The periodic signal is a sinusoidal function, 5sin(t). The mathematical model is(19)$$y_{m}=\left\{\begin{array}{c} y(t)\\ y(t)+5\sin(t) \end{array}\right.\;\begin{array}{c} t<t_{m}\\ t\geq t_{m} \end{array}$$

5.Multiplicative Fault: Due to drastic changes in the operating environment, the scale factor of the FOG changes, and the output signal of the FOG is multiplied by a coefficient from a certain point. The mathematical model is


(20)
$$y_{m}=\left\{\begin{array}{c} y(t)\\ qy(t) \end{array}\right.\;\begin{array}{c} t<t_{m}\\ t\geq t_{m} \end{array}$$


The mathematical models for the five faults are shown in Equations (16)–(20). The fault settings are as follows: c=10 °/h, k=0.0015 °/h/s, m=1.25 °/h, $t_{m}=10\;s$. According to the mathematical models of the above five kinds of faults, the fault simulation model is built in a MATLAB environment. Based on the signals under normal conditions, five kinds of fault signals are obtained through simulation, as shown in Figure 8b–f.

**Figure 8 micromachines-16-00073-f008:**
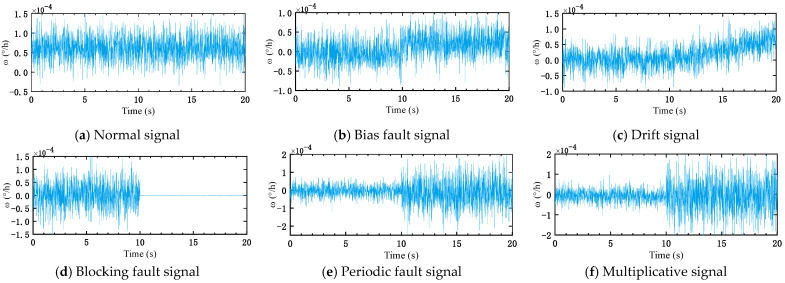
Fault categories of IFOGs, where (**a**) is the normal signal; (**b**) is the bias fault signal; (**c**) is the drift signal; (**d**) is the blocking fault signal; (**e**) is the periodic fault signal; (**f**) is the multiplicative signal.

### 2.3. Constructing Feature Vectors

IFOGs are susceptible to various types of noise interference in the vibration and shock environment [32,33]. The output signal contains highly informative high-frequency interference noise. Wavelet packet transform (WPT) completes the signal’s overall and local time domain analysis. Wavelet packet transform for the feature extraction of target signals has become very popular, e.g., Zhang et al. introduced the wavelet packet transform to extract feature signals from radar emission signals [34]; Huang et al. used wavelet packet transform to extract the energy features of the paint-stripping acoustic signals more efficiently [35]; Xia et al. used wavelet packet transform in combination with the energy spectrum for the processing of current signals of permanent magnet synchronous motors and obtained the main fault characteristics of the current signal [36]. According to the wavelet basis function requirements for the time-frequency domain signal support length, symmetry, vanishing distance order, and regularization, we select the db4 wavelet through several experimental validations and find that the three-layer wavelet packet decomposition of the fault signal is the most effective. The db4 wavelet is in the Daubechies family of wavelets, which are canonical orthogonal wavelets with tight support and a vanishing matrix of four [37]. Parseval’s theorem calculates the signal energy and wavelet packet reconstruction coefficients in Equations (21) and (22).(21)$${\left\|x(t)\right\|}^{2}={\int_{-\infty }^{\infty }\left|x(t)\right|}^{2}dt$$(22)$${\int_{-\infty }^{\infty }\left|x(t)\right|}^{2}dt={\sum_{m}\sum_{n}\left|h_{m,n}\right|}^{2}$$
where x(t) is the input signal, ${\left\|x(t)\right\|}^{2}$ is the signal energy, $h_{m,n}$ is the reconstruction factor, and m is the number of decomposition layers and n is the number of signals in the frequency band.

Since the signal energy and wavelet packet reconstruction coefficients satisfy the energy conservation relationship, we calculate the energy of each frequency band by reconstructing the coefficients, and we use the signal’s energy information as the feature vector.

In Figure 9, the original signal is the time-domain fault signal after wavelet packet reconstruction, and packet (3, 0)–packet (3, 7) are the signal components of the eight frequency bands in the three-layer.

In view of the significant differences in energy distribution among some frequency bands, it is necessary to normalize the feature vectors. The normalization formula is as follows:(23)$$e_{3,k}=\frac{{\left\|x_{3,k}(t)\right\|}^{2}}{\sum_{k=0}^{7}{\left\|x_{3,k}(t)\right\|}^{2}},k=0,1,\cdots \cdots 7$$
where $e_{3,k}$ is the normalized energy, and ${\left\|x_{3,k}(t)\right\|}^{2}$ is the energy in each band of the third layer.

After normalization, the energy distribution of different faults varies significantly, and the obtained feature vectors can represent the different operating states of the gyroscope well. Figure 10 demonstrates a set of energy distribution plots after normalization.

## 3. Design and Implementation of the NIGWO-iCaps NN Model

### 3.1. Improved Gray Wolf Optimization Algorithm (NIGWO)

Due to the disappearance or explosion of the gradient, we use the gray wolf optimization algorithm (GWO) to optimize the parameters of the enhanced capsule neural network (iCaps NN). This can eliminate the irreversible effects and avoid computational inefficiency. GWO is a meta-heuristic algorithm proposed by Mirjalili et al. in 2014 [38,39]. It has a simple structure, few parameters to adjust, a convergence factor that can be adaptively adjusted, and an information feedback mechanism. It performs better in terms of solution accuracy and convergence speed compared to other algorithms. Yue et al. [40] combined the gray wolf optimization and the fireworks algorithm to exploit the advantages of both algorithms and achieve the global optimum effectively. Zhang et al. [41] optimized the neural network algorithm by using the fast convergence property of the gray wolf optimization. They verified the superiority of the hybrid algorithm in terms of solution quality and computational efficiency.

GWO is an optimality-seeking algorithm that simulates gray wolves pursuing and encircling their prey. It mainly contains four steps: social hierarchy stratification, searching for prey, encircling prey, and attacking prey [37]. The wolf pack has a strict hierarchy, and gray wolves of GWO can adjust the convergence factor adaptively and provide information feedback promptly so that it can jump out of its limitations quickly and accurately when it falls into a local optimum. The mathematical model of the gray wolf predation stage is described as follows:(24)$$L=\left|C\cdot X_{p}(t)-X(t)\right|$$(25)$$X(t+1)=X_{p}(t)-A\cdot L$$
where t is the current iteration number, $X_{p}(t)$ is the current prey position vector, X(t) is the current gray wolf position vector, and C is the weight coefficient. A and L are the control and swing factors, respectively, which control the dynamic of the gray wolves’ movement.

The updating formulas of A and C are shown in Equations (26) and (27), where $r_{1}$ and $r_{2}$ are random scalars in the range of [0, 1]. a is the convergence factor, and its variation curve varies according to Equation (28). $t_{max}$ is the maximum number of iterations.(26)$$A=2a\cdot r_{1}-a$$(27)$$C=2\cdot r_{2}$$(28)$$a_{1}(t)=2(1-\frac{t}{t_{\max}})$$

Since the linearly decreasing convergence factor $a_{1}(t)$ cannot fully reproduce the actual hunting process of the gray wolf population, and the gray wolf population’s search range is too extensive in the early stage, it is very easy for the optimization to fall into local optimal or to find no optimal solution.

This article compares linear convergence factor $a_{1}(t)$, logarithmic convergence factor $a_{2}(t)$, exponential convergence factor $a_{3}(t)$, and tangent convergence factor $a_{4}(t)$. The models of these improved convergence factors are shown in Equations (29)–(31).(29)$$a_{2}(t)=\frac{2}{\mathrm{lg}2}\times {\left(\mathrm{lg}(2-{\left(\frac{t}{t_{\max}}\right)}^{2})\right)}^{3}$$(30)$$a_{3}(t)=a_{initial}+(a_{final}-a_{initial})\times {\left({\left(1-\frac{t}{t_{\max}}\right)}^{k_{1}}\right)}^{k_{2}}$$(31)$$a_{4}(t)=a_{initial}-(a_{initial}-a_{final})\times \tan(\frac{1}{\varepsilon }\cdot \frac{t}{t_{\max}}\pi )$$

The top three optimally adapted wolves in the population are α, β, and δ, and the other gray wolves update their positions based on the positions of the three top wolves until an optimal solution is obtained. By introducing the dynamic weighting strategy and changing the distribution of weights for positional updates within populations, GWO achieves better results. This paper uses proportional weights based on the step size Euclidean distance for position updating. The position update formula is shown in Equations (32) and (33).(32)$$W_{i}=\frac{\left|X_{i}\right|}{\left|X_{1}\right|+\left|X_{2}\right|+\left|X_{3}\right|},i=1,2,3$$(33)$$X(t+1)=\frac{1}{3}\sum_{i=1}^{3}X_{i}W_{i},i=1,2,3$$

Figure 11 shows the change curves of the four convergence methods. $a_{2}(t)$ is more in line with the demand. It decreases slower at the beginning of the iteration and maintains a more considerable value for longer, improving search efficiency. At the later stage, it decreases faster to maintain a smaller value for a longer time, which jumps out of the local optimum and enhances the search accuracy. $a_{2}(t)$ keeps as many parameters as possible within the optimal interval.

The pseudocode of NIGWO is shown in Algorithm 1 and can help people understand the implementation process more intuitively.
**Algorithm 1** NIGWO Algorithm1: Initialize gray wolf population V and maximum iterations Tm and population individual positions2: random generation parameter a, A, C3: Calculate fitness of population individual gray wolf4: while Stopping criterion not met do5:     for each gray wolf i do6:         Update position of gray wolf i by Equation (32) and (33)7:         Calculation parameter a by Equation (29)8:         Update parameter A and C9:     end for10:      Update dominance hierarchy11: end while

To test the performance of NIGWO, eight internationally used benchmark test functions are selected for validation tests, shown in Table 1, where $F_{1}∼F_{5}$ are the single-peak test functions and $F_{6}∼F_{8}$ are the multi-peak test functions. Jun Zhang et al. [42], Xuan Yanzhuang et al. [43], Başak H [44], Lidaighbi S et al. [45], Shaikh M S et al. [46], and Jiang J et al. [47] all used international standard benchmark functions to evaluate the performance of the algorithms.

In this paper, we select five optimization algorithms to compare and observe the optimization effect according to the fitness change. The five algorithms are the particle swarm optimization algorithm (PSO), the genetic algorithm (GA), the salp swarm algorithm (SSA), the gray wolf optimization algorithm (GWO), and the improved gray wolf optimization algorithm (NIGWO). The five algorithms set the same initial parameters. Among them, the particle velocity v is 3 in PSO, and the two learning factors are both 0.5; in GA, the crossover probability P_1_ is 0.5, the variance probability P_2_ is 0.85, and the selection probability coefficient is 5; and in SSA, the random number c_1_ is linearly reduced from 2 and decreases to 0, and c_2_ and c_3_ are both taken as 0.5.

Figure 12 compares the evolutionary results of five optimization algorithms, PSO, GWO, GA, SSA, and NIGWO, for eight test functions. Each algorithm was run independently 20 times. In Figure 12, NIGWO has the best convergence accuracy and convergence speed for the F_1_, F_4_, and F_8_ compared with the other four algorithms. It also has better convergence speed and accuracy than the other test functions.

To further verify the effectiveness of NIGWO, the mean value and the mean square deviation are chosen as the evaluation indexes. The results are shown in Table 2 and Table 3. MaxIter is the maximum number of iterations, where MaxIter is 300. Table 2 shows that NIGWO obtains reasonable global optimal solutions on all eight test functions and converges to the ideal value of 0 on the F_1_, F_6_, and F_8_ functions. For the other test functions, NIGWO has a better convergence accuracy compared to the other four algorithms, and from Table 3, the mean square deviation is 0 on F_2_ and F_4_, which has the best stability compared to the other four algorithms, NIGWO has a minor standard deviation on F_3_, F_5_, and F_7_. Combining the results of the two evaluation indexes, NIGWO has higher convergence accuracy and better stability.

### 3.2. Modeling of the iCaps NN

The inputs of Caps NN are involved in operations in the form of vectors. The structural framework of iCaps NN is based on Caps NN with the addition of the deep feature extraction module (DFE). DFE, the primary capsule layer, and the digital capsule layer form the encoder. The three fully connected layers form the decoder [19]. Its structural framework is shown in Figure 13.

First, DFE processed the data, and its structure is shown in Figure 14. To maximize the extraction of feature information, a small convolution kernel of 3 × 3 with Step 1 is used to convolve the 2D data matrix, followed by access to the maximum pooling layer with Step 2 for compression. Then, the data are passed to the 3 × 3 convolution layer with two layers of 256 channels, which outputs a feature map size of 2 × 2 × 256. Secondly, the extracted feature map is input to the primary capsule layer to convert scalar neurons to vector neurons. The primary capsule layer has 32 capsules, each with 16 dimensions, which are input to the digital capsule layer through dynamic routing iteration. The digital capsule layer has six capsules of length 32 and outputs six feature vectors finally, which correspond to the six types of faults. The vector modulus represents the probability of occurrence of the fault types. The network model parameters of iCaps NN are shown in Table 4.

The inter-capsule dynamic routing algorithm determines how active a feature in the primary capsule layer is in the digital capsule layer through the coupling coefficient $C_{ij}$. $C_{ij}$ is calculated by Equation (34) to obtain the input $s_{j}$ of the digital capsule layer.(34)$$s_{j}=\sum_{i}C_{ij}{u^{\prime }}_{j|i},{u^{\prime }}_{j|i}=W_{ij}u_{i}$$
where i = 1,……,32, j = 1, 2, 3, 4, 5, 6, and the product of the output $u_{i}$ of the primary capsule layer and the weight matrix $W_{ij}$ is weighted through $C_{ij}$ to obtain the input $s_{j}$ of the digital capsule layer. $C_{ij}$ is obtained by the iteration of inter-capsule dynamic routing and is calculated as shown in Equation (35).(35)$$C_{\mathrm{ij}}=\frac{\exp(b_{ij})}{\sum_{k}\exp(b_{ik})}$$(36)$$b_{ij}\leftarrow b_{ij}+{u^{\prime }}_{j|i}\cdot v_{j}$$(37)$$v_{j}=\frac{{\left\|s_{j}\right\|}^{2}}{1+{\left\|s_{j}\right\|}^{2}}\frac{s_{j}}{\left\|s_{j}\right\|}$$

The initial value of bias between the two capsule layers bij is 0, updated using Equation (36); the output vj of the digital capsule layer is obtained under the action of the nonlinear squeezing function Squashing in the capsule layer, calculated as in Equation (37). The pseudo-code implementation of the iCaps NN process is shown in Algorithm 2 to visualize the computation flow.
**Algorithm 2** iCapsule Neural Network Architecture1: Initialize input data X and labels Y2: Set number of capsules Nc3: Initialize primary capsules Ci with convolutional layers and hidden layer weight wL, L+1 and threshold value bL4: Sigmoid activation outputs of primary capsules5: for Routing (u |i,r,L) do6:     for each capsule i in layer L and capsule j in layer L+1: bij←0.7:     for r iterations do8:         For all capsule i in layer L: Cij ←softmax(bi) using Equation (35)9:         For all capsule j in layer (L+1) sj using Equation (34)10:          For all capsule j in layer (L+1): vj ←squash(sj) using Equation (37)11:          Update coupling coefficients bij using Equation (36)12:      end for 13: end for14: Compute output predictions vj


### 3.3. NIGWO-iCaps NN Implementation Process

The dynamic routing between capsules of iCaps NN is controlled by updating $C_{ij}$ to control the weight share of different capsules. However, it needs to be optimized using the Adam algorithm, which includes the other parameters of the whole network, the weight $W_{ij}$, and the threshold $b_{ij}$ of the hidden layer.

However, the learning rate ε, the number of iterations $N_{p}$ of the Adam algorithm, and the number of neurons D in the hidden layer of iCaps NN are determined by experts’ experience. To avoid the defects of artificially determining the model parameters, NIGWO is used to find the optimal D, $N_{p}$, and ε. Figure 15 shows the structural framework diagram of NIGWO-iCaps NN for the fault diagnosis of IFOG.

The steps for fault diagnosis of IFOG using NIGWO-iCaps NN are as follows.
Data AcquisitionThe state data for the six states are obtained and sampled using the IFOG simulation model. The sampling time is t = 20 s, and the sampling frequency is 100 HZ. The IFOG parameter settings: constant value drift is 0.1 °/h, random drift is 0.01 °/h;Data Pre-ProcessingSet 600 groups of each gyroscope signal for a total of 3600 signal samples. They will be feature extracted with WPT, and Shannon entropy is used to obtain the energy distribution of each group. After normalization, we obtain 3600 feature vectors: E_1_, E_2_, …, E_3600_, each matrix is 1 × 8 in size;Construction and Encoding of the 2D MatrixThe feature vectors are recorded into an 8 × 8 two-dimensional matrix Q by recording the fault data of IFOG in temporal and spatial order. This matrix is converted into a 2D matrix planar image H, as shown in Equation (30). Each element of the matrix is encoded.
(38)$$Q_{m}={\left[\;E_{8m-7},E_{8m-6},\ldots \ldots ,E_{8m}\right]}^{T},m=1,2,\ldots \ldots ,450$$
where $x_{ij}$ is the sample value of the i row and j column of the input matrix $Q_{m}$; $y_{ij}$ is the size of the output convolution value.Fault Feature Map ExtractionThe 2D matrix planar image H is input into NIGWO-iCaps NN, and after DFE, a 2 × 2 × 256 feature map is obtained. The obtained feature maps are reconstructed into 6 × 6 × 16 × 32 capsules to form the Primary Caps layer;Result OutputThe input is fed into the Digit Caps layer through the inter-capsular dynamic routing mechanism to obtain six feature vectors, and the magnitude of the length of the output vectors is the probability that the fault may occur. The output results are matched with the set fault data, and the diagnostic accuracy is obtained.

## 4. Experimentation

### 4.1. Experimental Environment Configuration

The experiments were conducted under the Win10 system, AMD Ryzen 5 3500U processor. The Pycharm platform configured the environment with Python 3.7.16, Tensor-flow 1.13.1, and NumPy 1.21.6; the software tools used were MATLAB 2021a, Jupyter 6.0.3.

### 4.2. Evaluation Indicators

This paper uses margin loss and accuracy as evaluation indexes to verify the model’s diagnostic performance for gyroscope faults. The formula for the loss value of each capsule vector is shown in Equation (39).(39)$$L_{c}=T_{c}\max{\left(0,\lambda_{1}-\left\|v_{c}\right\|\right)}^{2}+\rho (1-T_{c})\max{\left(1,\left\|v_{c}\right\|-\lambda_{2}\right)}^{2}$$
where c = 1, …, 6, the value of c represents the six states of the gyroscope; $T_{c}$ is the loss coefficient, when the predicted result corresponds to the actual result, $T_{c}$ = 1; $\left\|v_{c}\right\|$ represents the length of the capsule output vector, to a certain extent, and the magnitude of the fault probability; $\lambda_{1}$, $\lambda_{2}$ is the scale factor, $\lambda_{1}$ = 0.9, $\lambda_{2}$ = 0.1; the weight reduction factor ρ is 0.5. The total loss value of the model is shown in Equation (40), and the closer the value is to 0, the better the diagnostic performance of the model.(40)$$Loss=\sum_{c=1}^{6}L_{c}$$

The sample accuracy was calculated as follows:(41)$$Acc=\frac{1}{M}\sum_{i=1}^{M}\frac{\left|y_{i}\cap {\dot{y}}_{i}\right|}{y_{i}}$$
where ${\dot{y}}_{i}$ is the sample predicted label, $y_{i}$ is the actual label, and M = 3600 is the number of input samples. This formula represents the ratio of the number of correctly predicted labels to the number of true-for-correct labels, and the closer the value is to 1, the better the diagnostic performance of the model.

### 4.3. Determination of Model Parameters

Under the Griewank benchmark test function, NIGWO and the four optimization-seeking algorithms, PSO, GWO, GA, and SSA, are used to form a control test. Figure 16 shows the fitness curves of the five optimization-seeking algorithms for optimizing iCaps NN.

In Figure 16, NIGWO has the best convergence effect compared with the other four algorithms, reaching convergence at 13 iterations. NIGWO has a faster convergence speed and is closer to the optimal solution of the model parameters than GWO. That confirms the superiority of NIGWO in optimizing iCap NN for subsequent fault classification. In Figure 17, D, $N_{p}$, and ε eventually stabilize at 12, 759, and 0.0078, respectively.

## 5. Analysis and Discussion of Experimental Results

### 5.1. Stability Analysis Experiments

The dataset consists of 1920 samples, each containing 600 anomalies. The number of routing update iterations is taken as three. The initial learning rate factor of the Adam optimizer is 1.8 × 10^−3^, and the maximum number of iterations is 20. The simulation dataset is divided into training and test sets in a ratio of 4:1. Table 5 shows the fault labeling settings of IFOG. The simulation dataset is input into NIGWO-iCaps NN for testing, and the relationship between the loss value and the number of iterations of the model is shown in Figure 18, where (lg) is the logarithm of the number of iterations. With the increase in the number of iterations, the loss value gradually dropped to less than 2 × 10^−3^. This indicates that the misjudgment rate of the model’s diagnosis is gradually reduced to a controllable range.

To better see the prediction effect of NIGWO-Caps NN, this paper shows the diagnostic results for six gyroscope signal types using a confusion matrix. Label 1 denotes the normal signal of the gyroscope, and Labels 2 to 6 indicate the gyroscope bias faults, drift faults, blocking faults, periodic faults, and multiplicative faults, respectively. In Figure 19, the diagnostic accuracy of the model is 94.98% for periodic faults, 100% for normal signals, and more than 99% for the remaining four fault signals. It proves that the improved diagnostic model has an outstanding diagnostic performance in the paper.

### 5.2. Comparative Performance Analysis with Other Models

In Section 5.1, the diagnostic performance of NIGWO-iCaps NN is reasonable under the optimal values of the model parameters. We set up a comparison test to verify the influence of the optimized part of the model on the experimental results. Model I is the unoptimized Caps NN, Model II is the Caps NN optimized by the NIGWO, and Model III is the CNN with a similar structure to this paper. The primary and digital capsule layers are replaced with two fully connected layers. The number of neurons corresponds to the number of capsules, and the output neuron is six. Loss rates and classification accuracies under the simulated dataset are shown in Table 6.

In Table 6, after 200 iterations, Model III has the highest loss rate compared to the other three models, reaching 0.0186, and the accuracy rate is 83.44%. It has the lowest model diagnostic accuracy, which indicates that Caps NN is significantly better than the traditional CNN in terms of diagnostic performance. Comparing Model I and Model II, the loss of Caps NN optimized by NIGWO has dramatically reduced its loss value from 0.0364 to 0.0023. The classification accuracy has been substantially improved to 95.67%. Compared with Model II, the loss rate of NIGWO-iCaps NN reaches 0.0020, and the diagnostic accuracy reaches 96.95%, which is slightly better than Model II in terms of diagnostic performance.

The effect of DFE on classification accuracy is verified by increasing the number of iterations. Figure 20 shows the loss values of NIGWO-iCaps NN and Model II and the accuracies on the training and test sets after 500 iterations. The loss values of NIGWO-iCaps NN and Model II converge to 0.0017 and 0.0018 after 500 iterations, respectively. The difference is not too noticeable.

However, in Figure 21a, the accuracy of NIGWO-iCaps NN reaches more than 99.41% on both the training and test sets. In Figure 21b, the accuracy of Model I only reaches 85.53% on the test set, which is much lower than NIGWO-iCaps NN. It indicates that the ability to extract signal features and the deep spatial relationship between different features is greatly improved after adding DFE, significantly improving fault diagnosis accuracy.

To more intuitively observe the advantages of NIGWO-iCaps NN over Model II, Figure 22 shows the comparative histograms of the loss values of the two models after 500 iterations and the accuracies on the training and test sets. The loss value and accuracy of NIGWO-iCaps NN are better than those of Model II.

### 5.3. Generalization Performance Evaluation

To evaluate the robustness and generalization ability of NIGWO-iCaps NN to diagnose fault data output from gyroscopes in natural environments. In this paper, actual datasets from six different scenarios are selected for the performance evaluation of the diagnostic model. The actual datasets come from the PSINS database, and they are reproduced by the PSINS toolbox.241109. The normal signal is taken from a set of error-free UAV inertial navigation flight data. The bias fault signal is taken from the output dataset of the static SINS/GNSS combined navigation system. The drift fault is taken from the original MEMS-IMU dataset of the underground pipeline measurement. The blocking fault is taken from the static dataset of the laser gyroscope acquisition under the static base. The cyclic fault is taken from the output of the sports car dataset of the FOG-IMU/GPS tightly combined navigation system. The multiplicative fault is taken from the FOG’s detection of a section of the railroad track dataset set. The input of the fault diagnosis model in this paper is the angular velocity signal, which is also applicable to other IFOGs. The fault diagnosis of laser and micromachinery in this paper can prove the universality and reliability of the fault diagnosis model. Four hundred sets of each type of fault are selected as input datasets, and 3000 sampling points are taken from each dataset. Table 7 shows the relevant parameter settings for the actual fault dataset.

The 2400 sets of actual datasets were randomly divided by 8:2. A total of 8/10 were used to train the neural network, and 2/10 were used for testing. The training set learns the patterns and features of the signal through large amounts of data, constantly adjusting internal parameters to minimize prediction errors. A test set is different from a training set. Test sets are used to predict unlabeled data. The two datasets are independent of each other for fair evaluation. To ensure that the experimental results were not affected by random factors, the experiments were repeated 20 times, and we took the mean as the final experimental result. The number of neurons of iCaps NN in the hidden layer D, the number of iterations Np, and the learning rate ε were 12, 759, and 0.0078, respectively.

The Adam optimizer determined the other parameters. The training results are shown in Figure 23, where the loss value of the model in the actual dataset converges to 0.005 as the number of iterations increases, and the average diagnostic accuracy gradually converges to 95.42%.

Figure 24 represents the average accuracy of repeating the experiment 20 times for actual fault signals, with one to six indicating normal signals, bias faults, drift faults, blocking faults, periodic faults, and multiplicative faults, respectively. The values of the diagonal indicate the misdiagnosis rate of the model.

Figure 24 shows the average accuracy of NIGWO-iCaps NN for the six faults: 89.35%, 85.28%, 86.65%, 88.85%, 91.98%, and 90.76%, respectively. The gyroscope’s actual fault signal is slightly lower than the simulation signal in classification accuracy due to fault characteristics that are not obvious and scattered in distribution. The diagnostic accuracy of the bias fault is the lowest, but it can meet the practical needs in terms of the average accuracy rate. The generalization experiment can fully verify the strong classification ability and high diagnosis rate of the fault diagnosis model in this paper.

## 6. Conclusions

In this paper, we propose a NIGWO-iCaps NN model to solve the problem of FOG fault diagnosis in combined navigation systems. By adding DFE, more representative data features are obtained, which help the capsule layer for fault classification. By NIGWO in combination with the Adam algorithm, the limitations of artificially determining the model parameters are effectively avoided, and the diagnostic speed and stability of the model are improved.

The loss value of NIGWO-iCaps NN on the simulation dataset is reduced to 0.00197 after 200 iterations, and the accuracy rate reaches 96.95%. Compared with Model I, the diagnostic precision and accuracy of iCaps NN under the optimal parameters are significantly improved; compared with Model III, the loss value of the model in this paper is significantly reduced, indicating that iCaps NN has a better ability to grasp the profound relationship between fault signal features than CNN. After 200 iterations, the margin loss of model II is reduced to 0.00201. Its accuracy is comparable to the NIGWO-iCaps NN.

However, after 500 iterations, it is evident that although Model II performs well on the training set, the accuracy only reaches 85.53% on the test set. In the actual dataset, the performance of this model is good, and the diagnostic accuracy reaches 95.42%. In summary, the high confidence of NIGWO-iCaps NN on the simulation dataset is verified by contrast experiments, and the generalization performance of NIGWO-iCaps NN is verified using the actual dataset. This confirms that the model demonstrates high diagnostic accuracy and strong generalization ability.

## Figures and Tables

**Figure 1 micromachines-16-00073-f001:**
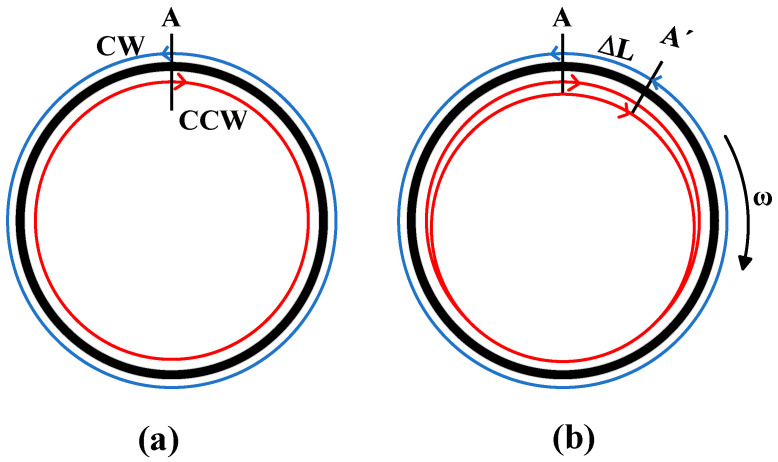
Schematic diagram of the Sagnac effect. (**a**) Static fiber optic ring (**b**) Rotating fiber optic ring.

**Figure 2 micromachines-16-00073-f002:**
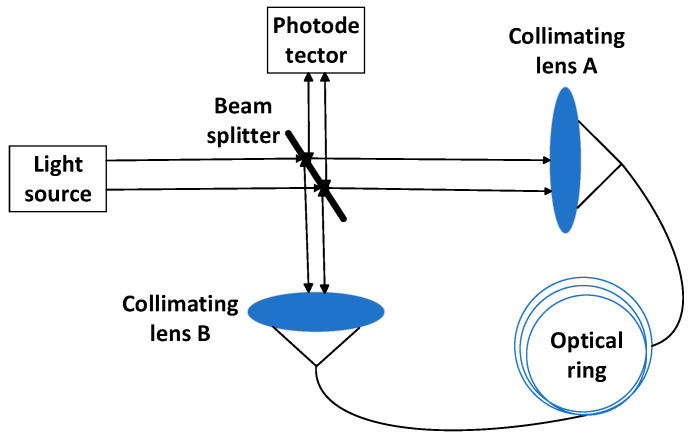
Basic optical path diagram of IFOGs.

**Figure 3 micromachines-16-00073-f003:**
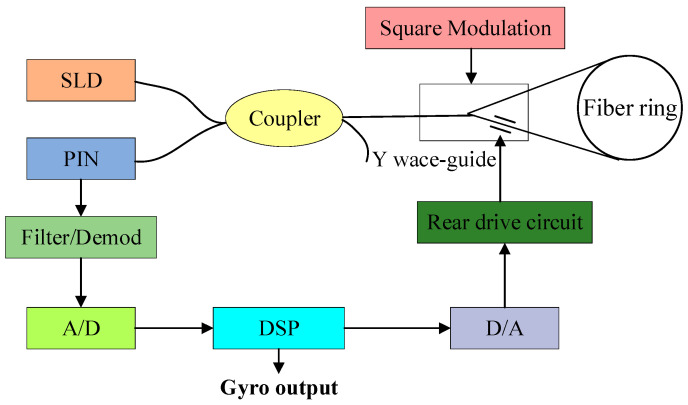
Physical model of a digital closed-loop fiber optic gyroscope.

**Figure 4 micromachines-16-00073-f004:**
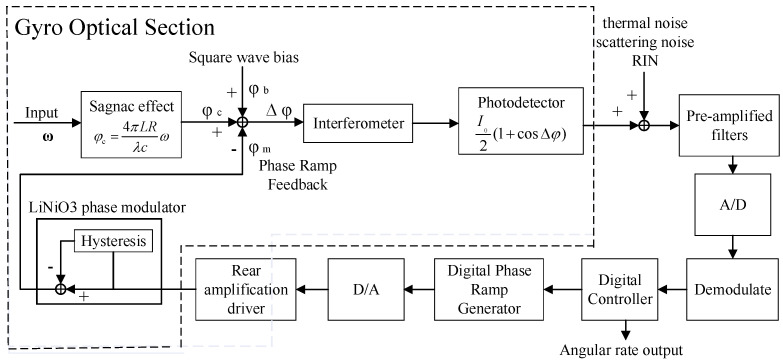
Block diagram of a digital closed-loop fiber optic gyroscope system.

**Figure 5 micromachines-16-00073-f005:**
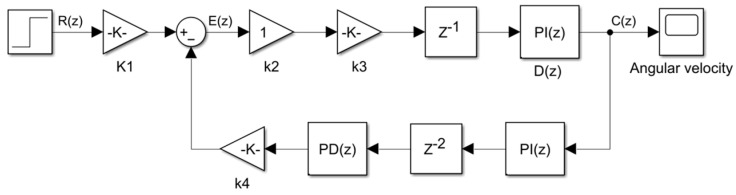
Simplified linear model of the digital closed-loop fiber optic gyroscope.

**Figure 6 micromachines-16-00073-f006:**
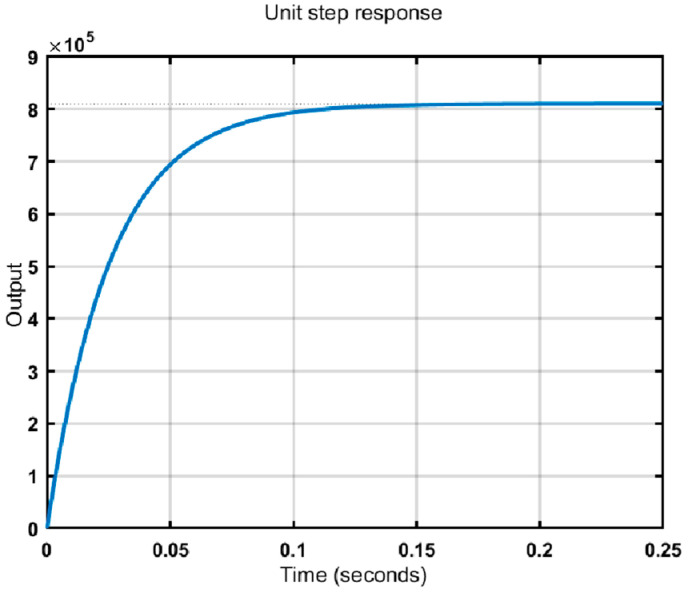
Unit step response curve.

**Figure 7 micromachines-16-00073-f007:**
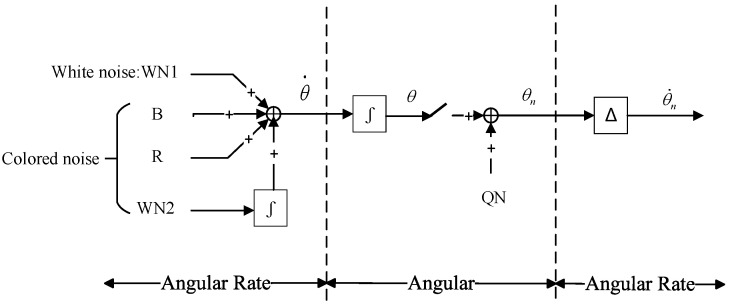
A stochastic model for digital closed-loop fiber optic gyros.

**Figure 9 micromachines-16-00073-f009:**
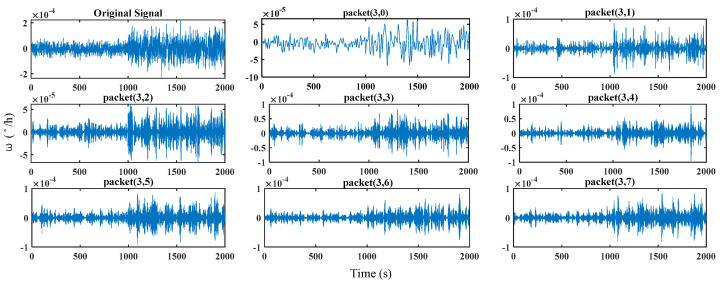
Signal components in eight frequency bands.

**Figure 10 micromachines-16-00073-f010:**
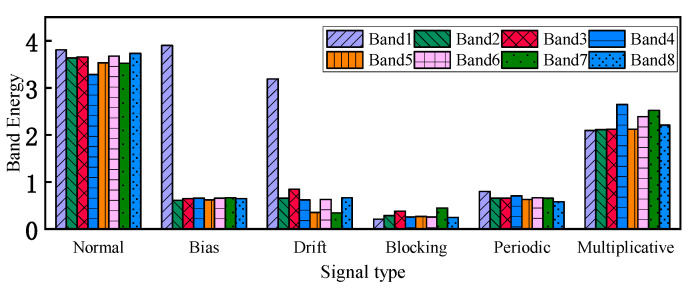
Histogram of normalized energy distribution.

**Figure 11 micromachines-16-00073-f011:**
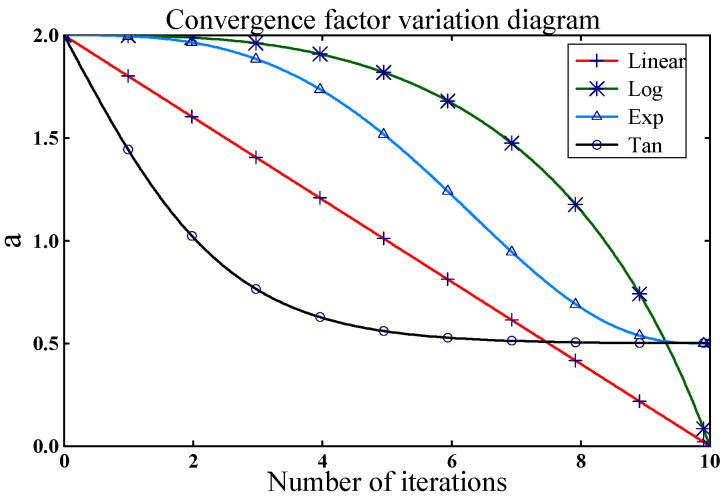
Convergence factor variation curve.

**Figure 12 micromachines-16-00073-f012:**
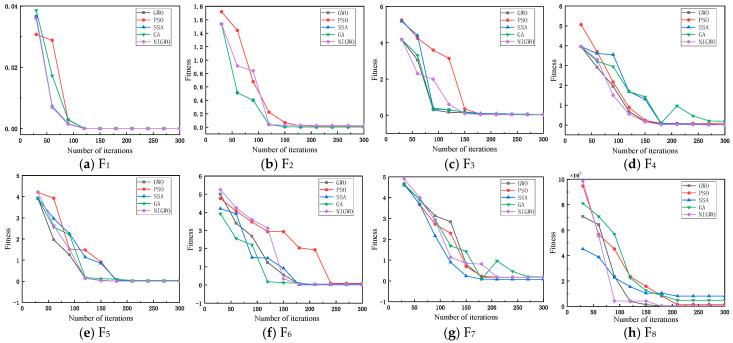
Convergence curves for the eight benchmarking functions.

**Figure 13 micromachines-16-00073-f013:**
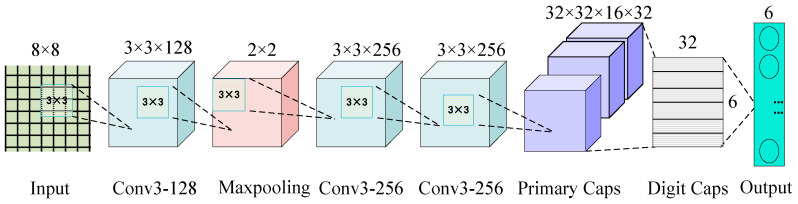
Schematic diagram of the iCaps NN structure.

**Figure 14 micromachines-16-00073-f014:**
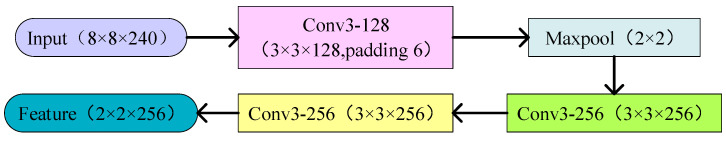
Deep feature extraction module (DFE).

**Figure 15 micromachines-16-00073-f015:**
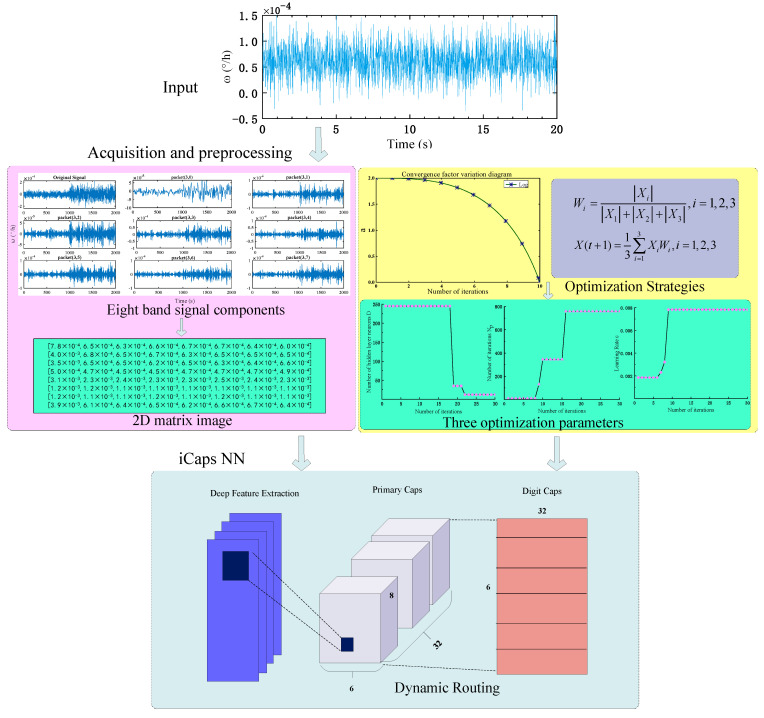
The general structural framework of NIGWO-iCaps NN.

**Figure 16 micromachines-16-00073-f016:**
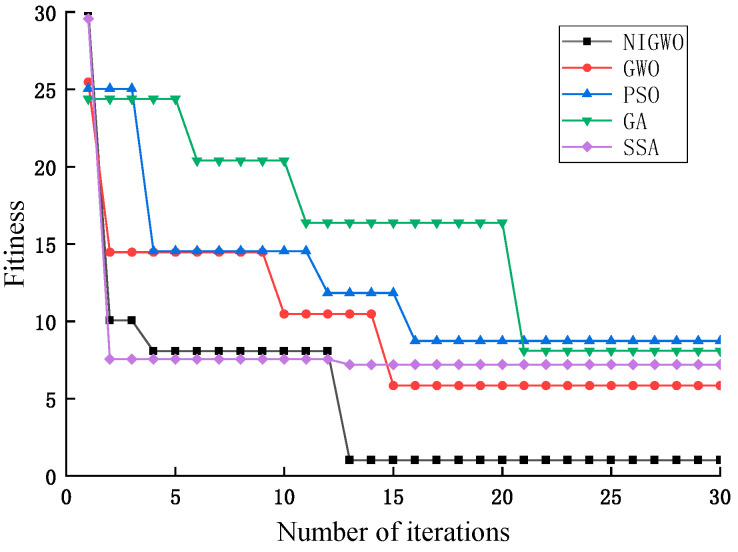
Comparison of changes in adaptation.

**Figure 17 micromachines-16-00073-f017:**
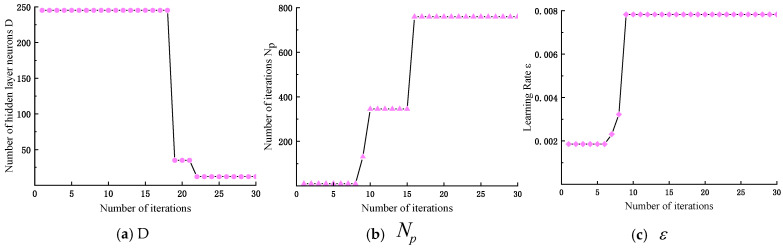
Variation of NIGWO-iCaps NN parameters.

**Figure 18 micromachines-16-00073-f018:**
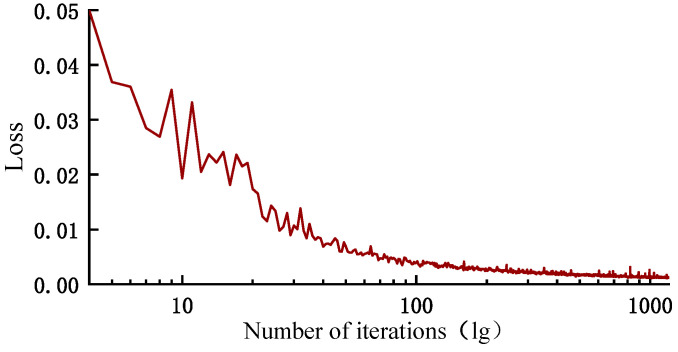
NIGWO-iCaps NN loss value curve.

**Figure 19 micromachines-16-00073-f019:**
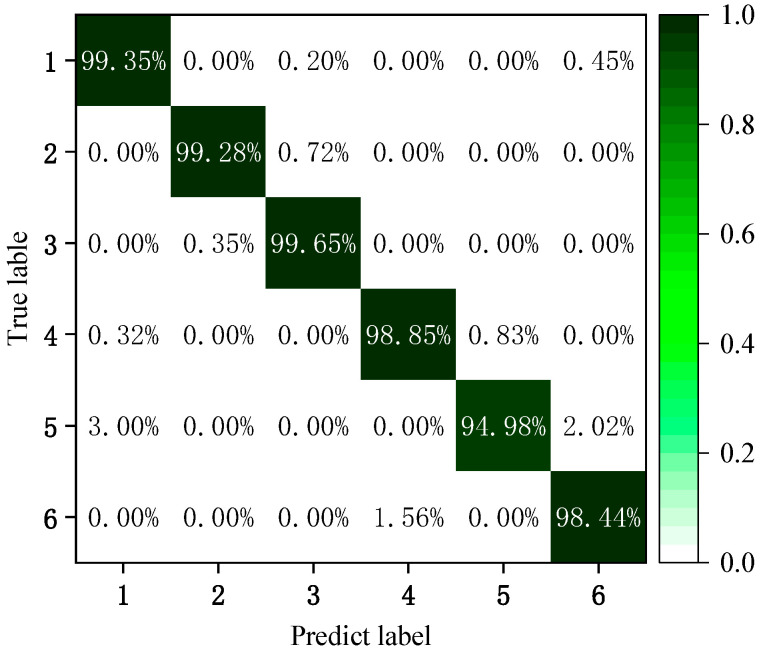
Gyroscope fault diagnosis results.

**Figure 20 micromachines-16-00073-f020:**
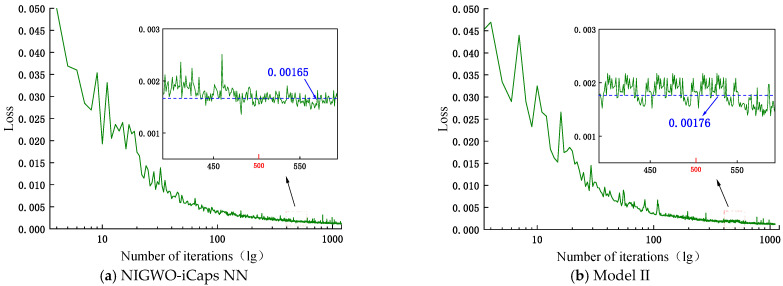
Loss value curves after 500 iterations for both models.

**Figure 21 micromachines-16-00073-f021:**
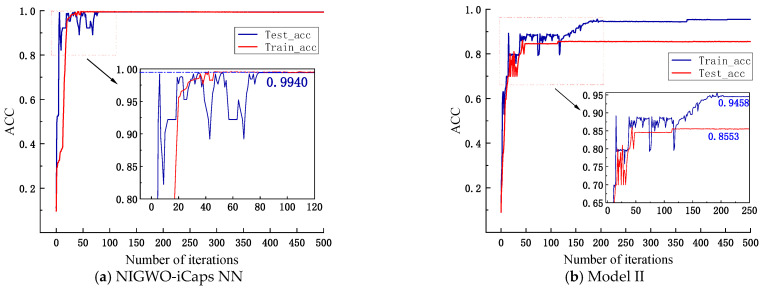
Accuracy curve after 500 iterations for both models.

**Figure 22 micromachines-16-00073-f022:**
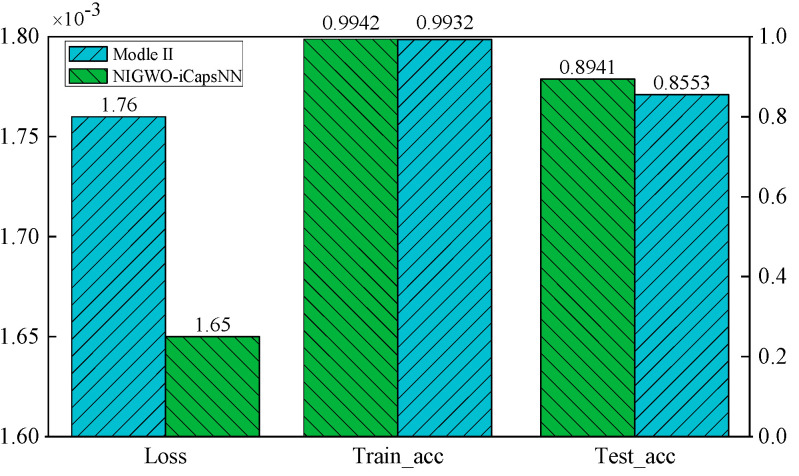
Comparison of loss values and accuracy between the NIGWO-iCaps NN and Model II.

**Figure 23 micromachines-16-00073-f023:**
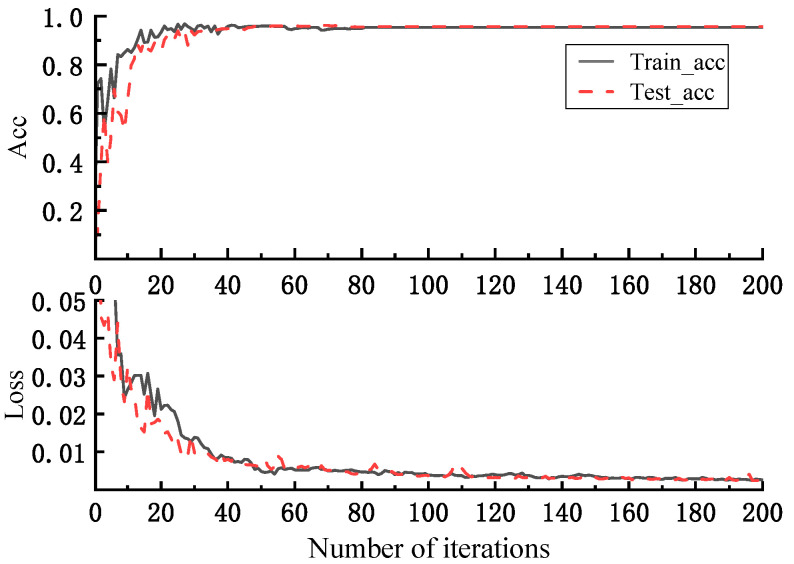
Loss value and accuracy curves for the actual dataset.

**Figure 24 micromachines-16-00073-f024:**
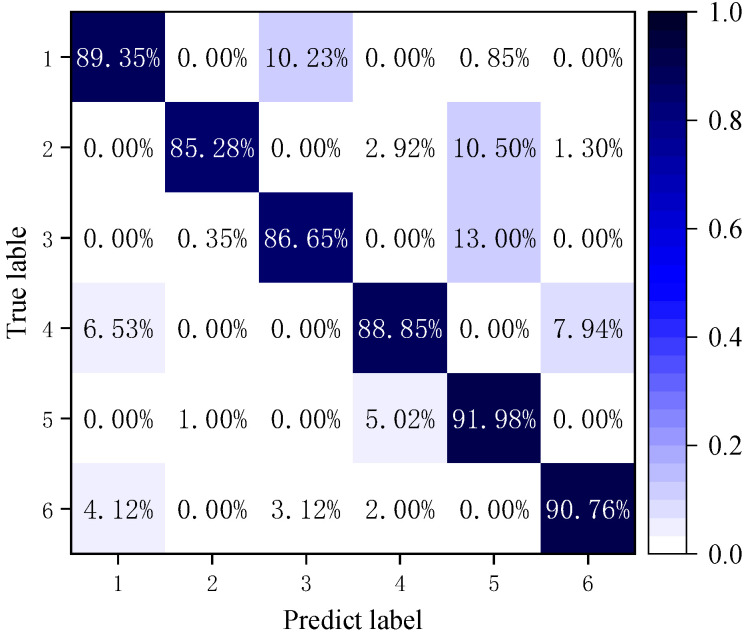
Classification accuracy for actual dataset.

**Table 1 micromachines-16-00073-t001:** Eight benchmarking functions and parameter settings.

Function	Function Expression	Dim	Hunting Range	Min
Sphere	$F_{1}(x)=\sum_{i=1}^{n}x_{i}^{2}$	10	[−50, 50]	0
Schwefel2.22	$F_{2}(x)=\sum_{i=1}^{n}x_{i}+\prod_{i=1}^{n}\left|x_{i}\right|$	10	[−10, 10]	0
Schwefel1.2	$F_{3}(x)={\sum_{i=1}^{n}\left(\sum_{j=1}^{i}x_{j}\right)}^{2}$	10	[−100, 100]	0
Schwefel2.21	$F_{4}(x)=\max_{i}\{\left|x_{i}\right|,1\leq i\leq n\}$	10	[−100, 100]	0
Quartic	$F_{5}(x)=\sum_{i=1}^{n}ix_{i}^{4}+random[0,1)$	10	[−1.5, 1.5]	0
Rastrigin	$F_{6}(x)=\sum_{i=1}^{n}\left[x_{i}^{2}-10\cos(2\pi x_{i})+10\right]$	10	[−10, 10]	0
Ackley	$\begin{array}{c} F_{7}(x)=-20\exp\left(-0.2\sqrt{\frac{1}{n}\sum_{i=1}^{n}x_{i}^{2}}\right)\;\;\;\\ \;\;-\exp\left(\sqrt{\frac{1}{n}\sum_{i=1}^{n}\cos(2\pi x_{i})}\right)+20+e \end{array}$	10	[−65, 65]	0
Griewank	$F_{8}(x)=\frac{1}{4000}\sum_{i=1}^{n}x_{i}^{2}-\prod_{i=1}^{n}\cos(\frac{x_{i}}{\sqrt{i}})+1$	10	[−50, 50]	0

**Table 2 micromachines-16-00073-t002:** Mean values of five algorithms with eight test functions.

Function	MaxIter	GWO	PSO	SSA	GA	NIGWO
F_1_	300	2.87 × 10^−15^	3.16 × 10^−22^	3.01 × 10^−15^	6.65 × 10^−14^	0
F_2_	300	6.91 × 10^−10^	3.52 × 10^−13^	2.79 × 10^−11^	3.51 × 10^−35^	3.49 × 10^−58^
F_3_	300	7.68 × 10^−8^	7.56 × 10^−18^	1.09 × 10^−9^	8.78 × 10^−16^	6.00 × 10^−27^
F_4_	300	9.80 × 10^−6^	9.44 × 10^−10^	9.10 × 10^−9^	9.86 × 10^−11^	9.00 × 10^−68^
F_5_	300	2.46 × 10^−5^	2.27 × 10^−3^	0	7.33 × 10^−8^	1.23 × 10^−12^
F_6_	300	2.21 × 10^−12^	3.78 × 10^−4^	6.75 × 10^−5^	2.31 × 10^−9^	0
F_7_	300	2.62 × 10^−6^	6.85 × 10^−5^	0	1.99 × 10^−10^	2.52 × 10^−36^
F_8_	300	0	5.51 × 10^−12^	2.56 × 10^−29^	4.89 × 10^−17^	0

**Table 3 micromachines-16-00073-t003:** Mean square deviation of five algorithms with eight test functions.

Function	MaxIter	GWO	PSO	SSA	GA	NIGWO
F_1_	300	9.85 × 10^−15^	5.94 × 10^−22^	5.28 × 10^−15^	8.93 × 10^−14^	0
F_2_	300	5.30 × 10^−10^	4.86 × 10^−13^	3.36 × 10^−11^	0	0
F_3_	300	3.34 × 10^−8^	2.57 × 10^−17^	2.10 × 10^−10^	2.57 × 10^−16^	8.89 × 10^−22^
F_4_	300	3.63 × 10^−6^	3.48 × 10^−9^	1.17 × 10^−9^	3.26 × 10^−12^	0
F5	300	1.81 × 10^−5^	4.63 × 10^−3^	0	9.97 × 10^−9^	6.64 × 10^−13^
F_6_	300	1.58 × 10^−12^	7.97 × 10^−4^	9.55 × 10^−6^	6.81 × 10^−10^	0
F_7_	300	1.44 × 10^−7^	7.43 × 10^−5^	0	9.94 × 10^−12^	3.22 × 10^−36^
F_8_	300	0	6.89 × 10^−12^	4.28 × 10^−29^	1.59 × 10−^17^	0

**Table 4 micromachines-16-00073-t004:** iCaps-NN model parameters.

Layer	Conv Kernel	Step	Padding	Output
Input	——	——	——	8 × 8 × 1 × 240
Conv 1	3 × 3	1	6	12 × 12 × 128 × 240
Pooling	2 × 2	2	——	6 × 6 × 128 × 240
Conv 2	3 × 3	1	——	4 × 4 × 256 × 240
Conv 3	3 × 3	1	——	2 × 2 × 256 × 240
Primary Caps	——	——	——	6 × 6 × 16 × 32
Digit Caps	——	——	——	32 × 6
Output	——	——	——	6

**Table 5 micromachines-16-00073-t005:** Gyroscope fault labels.

Signal Type	Normal Signal	Bias Fault	Drift Fault	Blocking Fault	Periodic Fault	Multiplicative Fault
Tag	000001	000010	000100	001000	010000	100000

**Table 6 micromachines-16-00073-t006:** Comparison of the diagnostic results of the four models.

Evaluation Index	Number of Iterations	Model I	Model II	Model III	NIGWO-iCaps NN
Loss	200	0.0036	0.0023	0.0186	0.0020
Acc/%		87.53	94.79	83.44	96.95

**Table 7 micromachines-16-00073-t007:** Parameter settings for the actual fault dataset.

Fault Type	Gyroscope Type	Gyroscope Acc (°/h)	SR ^1^ (HZ)	SL ^2^	Tag
Bias fault	FOG	0.5	100	3000	000001
Drift fault	MEMS	0.03	100	3000	000010
Blocking fault	Laser Gyroscope	0.01	100	3000	000100
Periodic fault	FOG	0.5	200	3000	001000
Multiplicative fault	FOG	0.05	200	3000	010000

^1^ SR is the sampling frequency, and ^2^ SL is the sampling length.

## Data Availability

The original contributions presented in the study are included in the article, further inquiries can be directed to the corresponding author.

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
