# Peer review of "NIGWO-iCaps NN: A Method for the Fault Diagnosis of Fiber Optic Gyroscopes Based on Capsule Neural Networks"

_micromachines, 2025, doi:10.3390/mi16010073_

Round 1

Reviewer 1 Report

Comments and Suggestions for Authors

In general, the manuscript “NIGWO-iCaps NN: a method for fault diagnosis of fiber optic gyroscopes based capsule neural networks" is suitable for publication in the journal Micromachines. The problem of diagnosing fiber optic gyroscope (FOG) malfunctions during system vibration is quite relevant. The proposed method allows recognizing five types of the faults, named by the authors: "Bias fault", “Drift fault", “Blocking fault", “Periodic fault", “Multiplicative fault". But the significance of the manuscript is reduced by the fact that the authors do not evaluate the values of the faults they have identified, do not diagnose the causes of these faults and do not determine which FOG elements are responsible for the appearance of these faults (for example, a Multiplicative fault may be caused by a malfunction of various FOG elements). In addition, it is necessary to raise the overall quality of the manuscript before publication.

Сomments:

1) In the abstract, the authors write “To minimize the impact due to vibration, this paper proposes a new fault diagnosis model, which is optimized by the enhanced capsule neural network (iCaps NN) optimized by the improved grey wolf algorithm (NIGWO) to diagnose the faults of the gyroscope.” But the article does not describe how and to what extent the proposed new fault diagnosis model minimizes the effect of vibration on FOG readings. You need to correct the annotation or add the appropriate text to the manuscript.

2) In the introduction. First of all, it is the use of graphical neural networks (i.e., “... avoids the negative consequences caused by maximum data aggregation and improves the ability of the model to extract features”). the main and secondary points.

3) It is obvious that the lack of the required concentration and mobility of FOG can be accompanied by the use of an adaptive Kalman filter. In the introduction, it is necessary to briefly describe how the problems highlighted in the manuscript are solved without the use of neural networks, and provide links to relevant sources.

4) Paragraph 2.1. It is necessary to provide a link to the source where you can find information about the work of IFOG. For example, “After modulation and demodulation of the signal, the angular velocity of the fiber gyroscope is obtained.”

5) Paragraph 2.1. An IFOG work option is described in the insufficient manner.

5.1) The authors write “... When the fiber optic ring rotates at an angular velocity omega, the two beams of light will produce a phase difference fi after completing a full rotational path, known as the Sagnac phase shift. After the photoelectric detector detects the phase difference signal, it outputs a voltage signal after the photoelectric conversion …”. It is not clear from the text how the phase difference is detected. In the manuscript, it is necessary to mention the interference of oncoming waves. –

5.2) The first paragraph of paragraph 2.1. should follow after the description of the IFOG operating principle. Otherwise, readers who are not familiar with IFOG will not understand it. –

5.3) The centense “The acquired angular velocity is integrated to obtain the output angular velocity of the fiber gyroscope.” it is not clear and may mislead the reader. Reformulate it.-

5.4) After expression 1, the parameter L is not signed in the text.

5.5) The block diagram in Figure 1 is not clear. It does not give an idea of the structure of the IFOG. I.e. it is not shown that there are two waves in the IFOG, how and where they separate and come together, where they are modulated in phase, etc. Either make changes to Figure 1., or add another figure with the IFOG scheme.

5.6) Text “At the same time, the angular rate is fed back to the input end through the subordinate LiNiO3 phase modulator, forming a closed-loop feedback system, and the output signal of the fiber optic gyro is obtained quickly and stably.” is not correct. It is not the angular velocity that is directed to the modulator, but the potential difference proportional to the angular velocity! As a result, the modulator creates a non-reciprocal phase shift, compensating for the shift caused by the Sagnac effect! You should also mention that the rectangular phase modulation is also created by the modulator (now it appears in Figure 1 out of nowhere).-

5.7) The text “The wavelength of the laser beam produced by the Superlight-emitting diode (SLD) is 1.5 mcm" is not correct. Strictly speaking, the Superlight-emitting diode radiation is not a laser (a diode is not a laser). –

5.8) Figure 4 is not sufficiently described in the text.

6) The authors write: “We select the db4 wavelet through several experimental validations and find that the 3-layer wavelet packet decomposition of the fault signal is the most effective.” You need to provide a link to the source, which describes the selected wavelet in detail.

7) Page 9. It is necessary to explain how the frequency bands were selected and why there are 8 of them.

8) Figure 7. In the figure, does “Normal” correspond to a serviceable IFOG? If so, it is not clear why the total energy of the signal (total over all frequency bands) in this case is greater than in the case of a “bias fault signal"? If this is related to rationing, then you need to briefly explain in the text how it was done.

9) Expression 18. You need to use ai instead of A.

10) Figures 9 and 13. Either redraw the algorithms in the form of flowcharts, or simply take them by surprise in the text (and not in the form of drawings).

11) Figure 11 is not sufficiently described in the text. Either describe it in more detail, or provide a link to the source.

12) A typo in the text: “In Figure 16, NIGWO has the best convergence effect compared with the other four algorithms, reaching convergence at 13 iterations...“ The link should be to Figure 15.

13) Paragraph 5.3. The article is devoted to IFOG, and experimental data are taken from different types of gyroscopes, including, laser and micromechanical. The justification of this technique must be proved in the text (to explain why this is permissible in this particular case).

14) Paragraph 5.3. Data for testing diagnostics of various Fault types are taken from various systems. It is necessary to explain why the diagnostics of all Fault types are not tested on the basis of experimental data from one system.

15) The authors write “The 2,400 sets of actual datasets were randomly divided by 8:2". It is necessary to explain in the text why the samples of experimental data were divided. 8/10 was used to train the neural network, 2/10 for testing?

16) The text of the manuscript does not describe how, when using experimental data, it was determined whether the Fault estimate given by the neural network was correct or not. Was data from other sensors (not IFOG) used?

Reviewer 2 Report

Comments and Suggestions for Authors

The paper deals with a neural network approach to identify faults in fiber optic gyroscope (FOG) inertial navigations systems. In general terms, the topic is expected to stimulate some interest, due to both the wide diffusion that FOG-based systems have in the present technological scenario and to the rather comprehensive discussion of the employed mehods and their implementation. In fact, although standard approaches are considered, combination of different tools reported in the paper can be of general interest also outside the specific problem treated in the paper.

Therefore, my opinion is that the paper can be considered for publication. Prior to acceptance, however, Authors are requested to consider the following criticisms from my side and revise the text, accordingly.

1.        The main drawback of the paper is its excessively technical content. I understand that, for instance, physics beyond FOG operation and related issues eventually leading to erroneous measurements are out of the main scope, but Authors must clarify several points such as:

a)        Actual origin of the Sagnac signal, that should essentially be represented by a beating frequency coming out from an interferogram, must be better clarified. Section 2.1 must be revised in order to better point out how the detecto can detect the phase difference signal and what is the physical meaning of the simulated signal.

b)        A careful revision must be operated on all equations, checking self-consistence of symbols: for instance, Eq. 1 contains a capital L symbol, that is not defined.

c)        Role of the “square wave bias signal”, appearing as a constant phase term in Eq. 2, must be clarified.

d)        The origin of the numerical values of K_1 – K_4 coefficients, see page 6, is unclear.

e)        Meaning of the “sinusoidal function 5sin(t)” right after Eq. 12, page 8, is unclear.

f)           The “db4” wavelet in Sec. 2.3, page 8, must be defined.

g)         I could not understand how the energy distribution of different fault mechanisms represent “well” the operation of the gyroscope, as stated in page 9, right above Fig. 7.

h)        The sentence “The top three optimally is too extensive in the early stage” must be rephrased, since cannot be understood.

i)            Benchmarking functions in Table 1 would deserve some comments and, mostly, adequate references.

j)            What are the “five algorithms” mentioned right after Table 1, at page 12?

k)         What’s the meaning of “(lg)” in the horizontal axis of Fig. 17?

l)           What’s the meaning of “especially 100% for normal signals” at page 19?

m)    Some more information would be useful on how the bias fault signal was retrieved in the experimental demonstration involving real, not synthetic, data (Sec. 5.3).

n)        Meaning of the sentence “with that before unoptimized” in the Conclusions is unclear.

2.        A very careful revision of language and style is strictly needed to make the presentation style adequate to a scientific publication. 

Comments on the Quality of English Language

A careful revision of the language and style all through the manuscript is needed. Authors are suggested to ask for a revision by a mother tongue reviewer.
